# Photoacoustic Properties of Polypyrrole Nanoparticles

**DOI:** 10.3390/nano11092457

**Published:** 2021-09-21

**Authors:** Peter Keša, Monika Paúrová, Michal Babič, Tomáš Heizer, Petr Matouš, Karolína Turnovcová, Dana Mareková, Luděk Šefc, Vít Herynek

**Affiliations:** 1Center for Advanced Preclinical Imaging (CAPI), First Faculty of Medicine, Charles University, 120 00 Prague, Czech Republic; peter.kesa@lf1.cuni.cz (P.K.); tomas.heizer@lf1.cuni.cz (T.H.); petr.matous@lf1.cuni.cz (P.M.); sefc@cesnet.cz (L.Š.); 2Institute of Macromolecular Chemistry, Czech Academy of Science, 162 06 Prague, Czech Republic; paurova@imc.cas.cz (M.P.); babic@imc.cas.cz (M.B.); 3Institute of Experimental Medicine, Czech Academy of Science, 142 20 Prague, Czech Republic; karolina.turnovcova@biomed.cas.cz (K.T.); marekova@biomed.cas.cz (D.M.); 4Second Faculty of Medicine, Charles University, 150 06 Prague, Czech Republic

**Keywords:** photoacoustic imaging, polypyrrole, nanoparticles, contrast agents

## Abstract

Photoacoustic imaging, an emerging modality, provides supplemental information to ultrasound imaging. We investigated the properties of polypyrrole nanoparticles, which considerably enhance contrast in photoacoustic images, in relation to the synthesis procedure and to their size. We prepared polypyrrole nanoparticles by water-based redox precipitation polymerization in the presence of ammonium persulphate (ratio *n*Py:*n*Oxi 1:0.5, 1:1, 1:2, 1:3, 1:5) or iron(III) chloride (*n*Py:*n*Oxi 1:2.3) acting as an oxidant. To stabilize growing nanoparticles, non-ionic polyvinylpyrrolidone was used. The nanoparticles were characterized and tested as a photoacoustic contrast agent in vitro on an imaging platform combining ultrasound and photoacoustic imaging. High photoacoustic signals were obtained with lower ratios of the oxidant (*n*Py:*n*APS ≥ 1:2), which corresponded to higher number of conjugated bonds in the polymer. The increasing portion of oxidized structures probably shifted the absorption spectra towards shorter wavelengths. A strong photoacoustic signal dependence on the nanoparticle size was revealed; the signal linearly increased with particle surface. Coated nanoparticles were also tested in vivo on a mouse model. To conclude, polypyrrole nanoparticles represent a promising contrast agent for photoacoustic imaging. Variations in the preparation result in varying photoacoustic properties related to their structure and allow to optimize the nanoparticles for in vivo imaging.

## 1. Introduction

Photoacoustic (PA) imaging (or optoacoustic imaging) is an emerging modality [1] asserting its place among non-invasive imaging methods in biomedicine and clinics [2,3]. The method is based on the photoacoustic effect described as early as in the 19th century [4]. When light interacts with a material (charged particles), a part of its energy is absorbed. The absorbed energy transformed into heat energy causes a thermal expansion and the subsequent formation of acoustic (sound) waves. Although almost any electromagnetic radiation may induce a PA effect, most applications require radiation in the ultraviolet to infrared wavelength range [5]. Excitation by near infrared (NIR) light (750–3000 nm) is preferred in in vivo imaging due to the higher tissue penetration of NIR light [6], because shorter wavelengths are absorbed or scattered to a greater extent. Both endogenous and exogenous chromophores may absorb excitation light and produce a PA effect. Naturally occurring endogenous chromophores are e.g., melanin, oxygenated or deoxygenated hemoglobin, however, their usage is often limited. Exogenous chromophores, or photoacoustic contrast agents, may substantially widen the range of potential PA imaging applications [7].

There are several possible approaches [8]. Molecular contrast agents [9], such as fluorescent dyes [10], can be used. A typical representative of dyes with absorption above 700 nm is the compound indocyanine green (ICG) [11]. Nanoparticles represent another approach [7,12,13]. In comparison with low-molecular dyes they broaden imaging ways due to their different biodistribution. In particular, nanoparticles leak significantly less into the interstitium; the signal loss is caused by organ or cell functions rather than plain diffusion, and nanoparticles can be better targeted to tissues or cells [14,15,16,17]. Particles based on noble metals usually reach absorption levels several orders of magnitude greater than traditional dyes, depending on their size, shape or surface [18]. Nanoparticles may be further modified and functionalized for use as multimodal [19,20,21] or theranostic agents [22,23,24,25].

Special attention is paid to nanoparticles formed by conjugated polymers, which may have optical properties that also make them suitable for PA imaging [24,26]. Conjugated polymers offer a larger conjugation length than polycyclic aromatic compounds, which leads both to an increase in the absorption coefficient and to a shift of absorption to higher wavelengths [27]. Compared to polycyclic aromatic compounds, these particles can also reasonably be expected to be less toxic, since conjugated polymers are practically insoluble in water and cannot, for example, intercalate with nucleic acids. Among others, nanoparticles based on polypyrrole have been studied [25,28,29,30,31].

It is reasonable to assume that, as with other nanoparticles, the method of preparation and product morphology influences the contrast properties and behavior of polypyrrole nanoparticles in biological systems. However, expert work on the application of polypyrrole particles in photoacoustics usually relies on a single preparation method and does not elaborate on the effects of individual particle parameters. Thus, it is not entirely clear how, for example, particle size, colloidal stability, degree of oxidation and surface coating affect contrast quality [32,33,34]. We have recently shown that by using different oxidants and surfactants/stabilizers at different concentrations, particle size and size distribution can be well controlled during synthesis at the same time [35,36].

Our aim was to test several synthesis procedures for preparation of polypyrrole (PPy) nanoparticles and to study the influence of modifications in polymer preparation method on their optical and thermophysical properties in vitro on a preclinical photoacoustic scanner, and, as a proof of principle, to test selected nanoparticles as a contrast agent also in vivo, to demonstrate their unique properties for preclinical, or even clinical photoacoustic imaging. Three different particle post-synthesis coatings were tested on cellular models.

## 2. Materials and Methods

### 2.1. Nanoparticle Synthesis

The nanoparticles (NPs) were synthesized by two different procedures. In general, the reactions were performed in septum-sealed flasks on a magnetic stirrer equipped with a thermoblock at the appropriate temperature. First, the specific amount of polyvinylpyrrolidone (PVP, *M*_n_ = 40,000, Sigma-Aldrich, Prague, Czech Republic) was dissolved in deionized water (18.2 MΩ.cm obtained from a water purification system Purelab Ultra, Evoqua Water Technologies, Guenzburg, Germany). Subsequently, the pyrrole monomer (Py, reagent grade 98%, purified by distillation in a vacuum under CaH_2_ and stored in a refrigerator at approximately 4 °C before use, Sigma-Aldrich) was added to the PVP solution and stirred for 15 min. After that, ammonium persulphate ((NH_4_)_2_S_2_O_8_, APS, reagent grade ≥ 98%, Sigma-Aldrich) or iron(III) chloride hexahydrate (FeCl_3_, reagent grade ≥ 98%, Sigma-Aldrich) solution was added dropwise to the reaction mixture under stirring at the appropriate temperature. To keep stable temperature during APS addition, the reactions were cooled down with ice bath at 5 °C for 10 min. The initially colorless systems gradually turned black in all cases and the polymerization was allowed to proceed for 24 h at the appropriate temperature. After polymerization, the PPy-NPs were washed by centrifugation with deionized water (APS: 8 times, for 1.5 h, 25 °C, 7830 rpm, FeCl_3_: 8 times, for 1 h, 25 °C, 11,500 rpm) and sonicated for 1 min.

Nanoparticles labeled as PPyA05, PPyA10, PPyA20, PPyA30, and PPyA50 were prepared by oxidation of Py monomer (300 mg, 4.42 mmol) with different molar ratios of APS (5 mL solution, *n*Py:*n*APS, 1:0.5, 1:1, 1:2, 1:3 and 1:5 respectively) in the presence of 1 wt.% PVP (65 mL) solution. Preparation was performed in the 100 mL septum-sealed flasks at room temperature [36].

Nanoparticles labeled as PPyF1, PPyF2, PPyF3, PPyF4, and PPyF5 were prepared by oxidation of Py monomer (1.0 g, 14.72 mmol) with FeCl_3_ (10 mL solution, *n*Py:*n*FeCl_3_, 1:2.3, 9.15 g, 33.85 mmol) in the presence of PVP solution of specific concentration (160 mL; 1, 1, 1, 0.6 and 0.1 wt.%, respectively). Preparation was performed in the 250 mL septum-sealed flasks at various temperatures (50 °C, 45 °C, 35 °C, 25 °C and 25 °C, respectively).

### 2.2. Physical Characterization of the Nanoparticles

The size, morphology and dispersity (*D*_n_ = Σ*n*_i_*D*_i_/Σ*n*_i_, *D*_w_ = Σ*n*_i_*D*_i_^4^/Σ*n*_i_*D*_i_^3^, *Đ* = *D*_w_/*D*_n_, where *D*_i_ is the diameter of particles, *n*_i_ is the number of particles, *Đ* is the dispersity, *D*_w_ and *D*_n_ are weight- and number-average particle diameter) were directly evaluated from microphotographs (using at least 300 particles) obtained by a transmission electron microscope (FEI-TEM, Tecnai G2 Spirit, FEI Company, Hillsboro, OR, USA) and analyzed with ImageJ analysis software [37]. The washed samples for the TEM analysis were prepared by depositing a diluted aqueous PPy-NPs dispersion on a grid with a copper membrane and carbon film and letting it dry at room temperature. Surface *ξ*-potential, hydrodynamic diameter (*D*_h_), and polydispersity index (*PI*) were measured by dynamic light scattering (DLS) using Zetasizer Nano Series ZEN3500 (Malvern, Worcestershire, UK) in disposable folded DTS1070 capillary cells in water. Measurements were performed three times. Arithmetic means ($\bar{x}$) and corrected standard deviations (s) of the mean of all the three parameters (*ξ*, *D*_h_, and *PI*) were calculated. The UV-Vis spectra of the aqueous dispersion of the washed particles were measured in physiological phosphate buffer (PBS saline tablet, Sigma-Aldrich) on a UV spectrophotometer Specord^®^ 210 Plus (Analytik Jena, Jena, Germany) using quartz cuvettes with an optical path of 1 cm and a wavelength range of 190–1100 nm.

### 2.3. In Vitro Photoacoustic Imaging

Suspensions of PPyA samples were diluted to concentrations 1, 2, 3, 4, 6, 8 mg/mL in a PBS solution (10 mM, pH = 7.4). PPyF samples were diluted to 0.2, 0.5 1, 2, 3, 4, 5 mg/mL (lower concentrations were selected in view of the anticipated in vivo tests).

Aliquots of the suspensions were injected into silicon tubes (inner/outer diameter 0.5/2.5 mm), adjusted into a phantom, and submerged in bubble-free water. The phantom was then scanned using a preclinical bimodal imaging platform Vevo 3100/LAZR-X (Fujifilm VisualSonics, Toronto, ON, Canada) combining high frequency ultrasound and photoacoustics imaging. Photoacoustic spectra were acquired with an MX400 transducer (20–46 MHz, 50 µm axial resolution) equipped with an original jacket for inserting the optical fiber bundle (14 mm). Photoacoustic spectra were measured in two wavelength ranges (680–970 nm and 1200–2000 nm, increment 5 nm; the used instrument enabled tuning of the excitation lasers in these two ranges only). Laser pulse energy was 36 mJ. Spectra in both ranges was obtained for each aliquot and each suspension batch. The spectra were evaluated by the dedicated software Vevo LAB (Fujifilm VisualSonics). Experimental setup is shown in Appendix A.

### 2.4. Nanoparticle Coating

PPyF3 nanoparticles were coated to improve their biocompatibility for future in vivo applications. Three different coatings (poly-l-lysine PLL (*M*_w_ ~70000, Sigma-Aldrich), Pluronic F-127 (Sigma-Aldrich), and diblock DB3) were used for coating of PPy-NPs by following procedures:(1)An aqueous solution of PLL or Pluronic F-127 (0.2 mL, 30 mg/mL) filtered through a sterile syringe filter (polyvinylidene difluoride membrane (PVDF), 220 nm) was dissolved in 0.985 mL of water (Aqua pro injectione) and consequently 1.015 mL of PPy sterilized colloid (19.7 mg/mL) was added dropwise under sonication (1 mm tip diameter; 20 W) and finally sonicated for 2 min;(2)PPy sterilized colloid (1.015 mL, 19.7 mg/mL) was diluted with 1.985 mL of water (Aqua pro injectione) and 0.2 mL of filtered (PVDF, 220 nm) poly(*N*,*N*-dimethylacrylamide)-*b*-poly(2,2,2-trifluoroethyl acrylate), *M*_w_ = 10-*b*-10 kDa (abrev. pDMAAm-*b*-pTFEA; DB3) solution in acetone was added dropwise under sonication, sonicated for 2 min and finally reduced in septum sealed ampule on vacuo to volume 2 mL. pDMAAm was chosen for its previously reported suitability for particle biocompatible coating [38] and pTFEA was chosen due to its lipophilic character and light transparency. The main suggested coating mechanism is built on absorption of pTFEA block onto PPy surface, while water soluble pDMMAm block floats in dispersion medium and acts as steric stabilizer.

Diblock copolymer pDMAAm-*b*-pTFEA, *M*_w_ = 10-*b*-10 kDa, was synthesized by two-step solution reversible addition−fragmentation chain-transfer (RAFT) polymerization. 12 g (121 mmol) of *N*,*N*-dimethylacrylamide was dissolved in 30 g of dried ethyl acetate together with 0.4375 g (1.2 mmol) of 2-(dodecylthiocarbonothioylthio)-2-methylpropanoic acid and 33.31 mg of 4,4′-azobis(4-cyanovaleric acid) (ACVA). The solution was purged with argon (10 min) and polymerized at 70 °C for 65 min in septum-sealed vial. The synthesized pDMAAm was triple precipitated with diethylether. Its molecular weight was confirmed by gel permeation chromatography (GPC) equipped with DLS detector and the functionality (presence of active transfer groups) was measured with UV-Vis. Consequently, 2 g of pDMAAm (0.2 mmol; 95% functionality) was dissolved in 20 g of dried ethylacetate together with 6 mg (0.02 mmol) of ACVA and 1.770 g (13.9 mmol) of 2,2,2-trifluoroethyl acrylate and polymerized under the conditions described above for 24 h. The diblock was triple precipitated with cold *n*-hexane (selective solvent of pTFEA) and the final molecular weight was confirmed gravimetrically. The deactivation of the transfer groups was done with molar twentyfold of azobisisobutyronitrile in dried ethyl acetate at 70 °C. All chemicals were purchased from Sigma-Aldrich.

### 2.5. Cell Proliferation in the Presence of the Nanoparticles

The proliferation of C6 cells in the presence of both coated and uncoated PPyF3 particles was tested. Cell proliferation curves were obtained using the xCELLigence^®^ RTCA DP instrument (ACEA Biosciences, San Diego, CA, USA). The method is based on the measurement of electrical impedance on a special 16-well microplate (E-Plate) equipped with gold electrodes at the bottom. As the cells grow, the impedance increases (adherent cells impede the flow of charge carriers in the cultivation media, which serves as electrolyte. The actual impedance depends on cell shape and size, the number of cells and the attachment quality. Impedance is expressed as the Cell Index (dimensionless parameter). Fifty μL of cultivation media was added to each well and the background impedance measured. Next, C6 cells (10,000 cells per well) were seeded and left to attach for 30 min. During the log phase (approximately 2 h after the beginning of the experiment), the nanoparticles were added to achieve a final concentration of 1, 0.5, 0.25 mg/mL in the well. Cells were then cultivated under standard cultivation conditions. Impedance was recorded every 15 min for 48 h. Each experiment was performed in doublets.

### 2.6. In Vivo High-Frequency Ultrasound and Photoacoustic Imaging

For in vivo experiments, PPyF4 nanoparticles with PLL coating were used. The nanoparticles were tested in vitro by an MTT cytotoxicity/proliferation assay [39] prior to in vivo application. The MTT assay is a qualitative colorimetric method based on the metabolic reduction of soluble thiazol-tetrazolium bromide (MTT) to insoluble formazan with an absorption maximum at 570 nm.

The fibroblast cell line (normal human dermal fibroblasts—NHDF) was cultured with DMEM media (Sigma-Aldrich) supplemented with 10% fetal bovine serum (Sigma-Aldrich) and 1% penicillin-streptomycin antibiotics cocktail (Biosera; Nuaille, France) in a 5% CO_2_ humidified atmosphere, cultured once a week.

One day prior to application, 10,000 cells/cm^2^ were seeded into the 96 wells plate. The empty medium was used as a blank. Nanoparticles were added into the wells at concentrations 10; 30; 70; 150; 300; 600 or 1000 µg/mL and cultured for 24 h or 48 h. Cells with PBS addition were used as a control and treated in the same way. Cells were thereafter washed with pre-warmed PBS twice and filed with 100 µL PBS + 100 µL 10% SDS. Absorbance was measured using a Spark multimode microplate reader (Tecan, Männedorf, Switzerland) after the overnight cultivation with the solvent on the 570 nm wavelength; the 690 nm wavelength was used as a reference.

Multimodal in vivo imaging with PPy NPs was performed on 12–14 weeks old Nu/Nu mice (*n* = 3, males) bred in a specific-pathogen free animal facility of the Center for Experimental Biomodels, First Faculty of Medicine, Charles University (Prague, Czech Republic). Animals were kept in individually ventilated cages (12/12 h light/dark cycle, 22 ± 1 °C, 60 ± 5% humidity) with free access to water and a standard rodent diet.

The mice were anesthetized by spontaneous inhalation of isoflurane (3% for induction and 1.5–2% for maintenance) at 1.2 L/min air flow during the experiment. A cannula was inserting into the tail vein using a 30 G painless steel needle (Transcodent GmbH&Co. KG, Kiel, Germany), then the mice were positioned onto a heated working table (Fujifilm VisualSonics, Inc.) in a supine position and conductively connected by four electrodes to monitor ECG and breathing. The temperature was set to 37.7 °C. The position of the Mx400 ultrasound transducer equipped with a jacket with inserted a narrow (14 mm) optical fiber bundle (Fujifilm VisualSonics, Inc.) was adjusted above the animal to achieve the best parasternal long axis view of heart’s left ventricle, aorta, left and right atriums. The distance between transducer surface and animal skin (5 mm) was filled with a bubble free transparent ultrasound gel (OXD, Spain). For the imaging study, The Mouse (Large) Cardiology Preset was chosen. The total image depth was set to 17 mm in all scans. A total 2D Gain was set to 29 dB and total PA Gain was adjusted to 40 dB. Low persistence was allowed in all records. The Time Gain Compensation (50-55-55-55-55 dB) was used to obtain a more intensive PA signal from deeper tissue layers. Nanoparticles (20 µg of PPYF4 NPs per 1 g of animal weight) were diluted in 10 mM PBS (pH 7.4) and administrated intravenously through the cannula by Vevo Infusion Pump (Fujifilm VisualSonics, Inc.) in a total volume of 180 µL within 4.2 s. PA signal was recorded continuously in the mouse heart at 800 nm (laser pulse energy 30 ± 1 mJ, pulse rate 20 Hz, pulse width < 10 ns) before, during, and for approximately 9 min post administration. 800 nm was selected to minimize nature background.

The experimental setup is shown in Appendix A.

Post-processing of the obtained data was realized using VevoLAB software V.3.2.5 (Fujifilm VisualSonics, Inc.). To decrease the natural PA signal originated from the hemoglobin an initial value of PA signal measured before NPs administration as a single wavelength scan at 800 nm was subtracted using a Subtraction Control function in all scans. Then, exported numerical data were analyzed using the OriginPro 8 software (OriginLab Corporation, Northampton, MA, USA). A One-Way ANOVA and Tukey Test were used to calculate a p-value (statistically significant p-value was assessed as a *p* < 0.05).

Photoacoustic spectrum of PPyF4 nanoparticles (used for in vivo experiment) was compared to the spectrum of commercially available indocyanine green (ICG) at the same concentration of 1 mg/mL.

The animal experiments were performed in accordance with national and international guidelines for laboratory animal care and were approved by the Laboratory Animal Care and Use Committee of the First Faculty of Medicine, Charles University, and the Ministry of Education, Youth and Sports of the Czech Republic (MSMT 7025/2018-2).

## 3. Results

### 3.1. Chemical and Physical Characteristics

The presented PPy-NPs were prepared by two different experimental methods in the presence of PVP which was used as a polymer stabilizing particle during the polymerization. The absence of PVP vibration bands in the Raman spectra confirmed that the PVP molecules are present solely in the particle matrices rather than on the surfaces, which offers a possibility for post synthetic modification in future stages [36]. The effect of different APS molar ratio on the particle size, hydrodynamic properties, morphology and UV-Vis and photoacoustic properties was examined. The use of PVP during the polymerization allowed controlling the particle size and polydispersity. The *D*_h_ of the particles increased with the increasing molar ratio of APS from 200–500 nm, as did also *PI*. The ζ-potential of particles suspended in distilled water was found in the range −8 to +2 mV (pH ~3, 25 °C), see Table 1. The TEM results showed that the morphology of PPyA05–PPyA50 changed from a spherical to a slightly elliptical shape with increasing APS molar ratio (Figure 1). The non-spherical shape is probably caused by overoxidation due to strong oxidation activity of APS. In addition, the overoxidation softens particle surfaces, which causes a coalescence of particles (sintering) due to the capillary forces that occur during particles drying on the TEM grid deposition. Furthermore, the *D*_n_ of PPyA05-PPyA50 was determined in the 38–50 nm range with a narrow particle size distribution and *Đ* (see Table 1). Ultimately, the aforementioned overoxidation changes even the UV-Vis spectra of the nanoparticles formed. The absorption band in the 600–1100 nm range in the near infrared region with a maximum at approximately 900 nm is attributed to the transition from the valence band to the bipolaron band (PPyA05, PPyA10, PPyA20) while the bipolaron absorption band of the PPyA30 and PPyA50 gradually reduces with increasing APS molar ratio (Figure 2).

### 3.2. In Vitro Photoacoustics

Photoacoustic spectra of the PPyA nanoparticles prepared by polymerization at different APS molar ratios are shown in Figure 3. Samples with a lower ratio of the oxidant (PPy05, PPy10, PPy20) revealed a substantially higher signal compared to the samples with a higher ratio of the oxidant (PPy30, PPy50). The effect was noticeable in both wavelength ranges (680–970 nm, 1200–2000 nm). Interestingly, the effect of the used APS amount is not linear, and the highest signal was obtained at Py:APS ratio 1:1.

The influence of nanoparticle size was studied on the PPy nanoparticles prepared with FeCl_3_. Nanoparticle sizes varied from 48 to 131 nm (see Table 1). Figure 4 shows the dependence of the relative PA signal (recalculated per particle) on nanoparticle size. As the dependence was observed to be nonlinear (see Figure 5a), a plot of the PA signal vs. particle cross section is also provided (Figure 5b).

### 3.3. Cell Proliferation in the Presence of the Nanoparticles

Proliferation curves are presented in Appendix A. Both coated and uncoated PPyF3 nanoparticles transiently slowed down C6 cell proliferation at all tested concentrations. In the long term, cells in the presence of low concentration (0.25 mg/mL or 0.5 mg/mL) of uncoated and PLL-coated nanoparticles recovered and started to proliferate, while higher concentration (1 mg/mL) led to impairment of proliferation (see Appendix A). Pluronic-127 and DB3 coatings turned out to be unsuitable; cell proliferation decreased at all concentrations of Plu-127-coated nanoparticles, and at 0.5 and 1 mg/mL concentrations of DB3-coated PPyF3 nanoparticles (Appendix A).

### 3.4. In Vivo Imaging

We utilized PPL-coated PPyF4 nanoparticles with a 78.2 nm diameter for in vivo photoacoustic imaging. A cytotoxicity test performed on NHDF cells prior to the in vivo experiment revealed low cytotoxicity of the nanoparticles. Cell viability was 87 ± 6% after 24-h incubation, and 74 ± 5% after 48 h at the highest concentration (1000 µg/mL).

The nanoparticles were injected into the tail vein and the photoacoustic signal was acquired at 800 nm. The intense photoacoustic signal was clearly visible in the anterior wall of the mouse heart immediately after NPs injection (Figure 6b). Lower intensities of PA signal were also detected from the area of the aorta and both atriums. The PA signal generated by the nanoparticles reached its maximum approximately 5 s post injection, then slowly decreased ((Appendix A). The equilibrium between blood and nanoparticles occurs approximately after 8 min when PA signal detected at 800 nm reached the reference value.

PPyF4 nanoparticles provided higher signal in the whole excitation range of 680–970 nm than commercial indocyanine green, both spectra are shown in Appendix A.

## 4. Discussion

Photoacoustic imaging is an interesting technique for both native and contrast enhanced imaging, which has already established its position among other imaging methods in biomedicine [40]. Its applications depend on the knowledge of contrast mechanisms and contrast agent properties. Our study focused on polypyrrole nanoparticles, reputed for their interesting optical and photoacoustical properties. Nonetheless, relevant information is still scarce.

We tested particles prepared by different methods and compared variations in their preparation. Polymerization with different amounts of ammonium persulphate as an oxidative agent yielded samples with strikingly different PA properties. Low APS ratio led to particles with a high PA signal, whereas at higher APS ratios, particles with a low PA signal were formed. The PA signal probably corresponds to the portion of conjugated bonds in the polymer, which is higher at low ratio of the oxidant. High oxidant ratios led to disruption of conjugation (both on short and long distance) by increasing the number of over-oxidized structures. This probably caused a shift of the absorption band in the spectra to shorter wavelengths and the PA signal to decrease at the 680–970 nm wavelength range.

Particle size is important from two perspectives. Firstly, a selected size determines the NP biodistribution, and secondly, the particle size strongly influences signal intensity. If the optical skin depth of the nanoparticles is smaller than the core size (which may apply in the diameter range we used in our experiment), heat absorption becomes proportional to the particle cross section instead [41]. Then, a linear dependence of PA signal on particle cross section would be anticipated. Although the limited number of samples characterized by different sizes did not enable a reliable evaluation, the data obtained (Figure 5b) tend to indicate a linear dependence of the signal on the particle cross section. The measured dependence also proved, that smaller particles (at the same monomer concentration) are more efficient than bigger ones.

Cytotoxicity of the nanoparticles strongly depended on concentration and coating did not improve nanoparticle safety. However, the safe concentrations of uncoated or PLL-coated NPs were still substantially higher than anticipated concentration in potential in vivo applications. Interestingly, while PLL coating was reported as a suitable coating for iron oxide nanoparticles [42] improving nanoparticle biocompatibility, no substantial change was brought by PLL coating of polypyrrole nanoparticles: Both PLL-coated and uncoated particles revealed similar proliferation index at all tested nanoparticle concentrations.

Medium-sized particles (PPyF4, diameter 78.2 nm) were selected for the in vivo experiment. They were anticipated to be biocompatible (size below 100 nm), but still big enough to provide a sufficient PA signal. Performed tests confirmed their biocompatibility. The subsequent in vivo experiment proved that the nanoparticles could be easily visualized in an experimental mouse. We successfully detected a high PA signal of the nanoparticles in the heart anterior wall. A significantly lower PA signal can be also detected in the left ventricle, aorta and both right and left atriums due to the high blood flow. The light penetration depth and the fact that the light beam was focused from the top may also influence the resulting photoacoustic intensity.

We also confirmed that polypyrrole nanoparticles provided a substantially higher PA signal at the same experimental conditions than a commercial ICG, which is broadly used as a photoacoustic tracer or label [43,44,45]. The intent of the study was indeed the physical characterization of the particles, which may enable further optimization of their preparation for future applications. Therefore, the in vivo experiment was intended as a proof of concept only, a detailed preclinical in vivo evaluation of PPy nanoparticles including extensive toxicity testing, biodistribution, excretion pathways, etc., still needs to be carried out. Even though, the experiment outlined possible application of tailored polypyrrole nanoparticles in e.g., cardiologic diagnostic imaging. Moreover, future optimization of PPy nanoparticles in terms of maximizing the signal at the required size and surface modifications may enable specific targeting, e.g., of tumors, and possible utilization for photothermal therapy [25,30,33,46].

## 5. Conclusions

We synthetized and tested polypyrrole nanoparticles as a promising substance for photoacoustic imaging. Variations in their preparation methods led to different photoacoustic properties corresponding to nanoparticle chemical structure and size. Low oxidant ratio during polymerization leading to higher number of conjugated bonds increased PA signal of the nanoparticles, and vice versa. Particle size also strongly influenced signal intensity. While heat absorption depends rather on the particle volume, a linear dependence of PA signal on particle cross section corresponds to an anticipated surface-dependent absorption. It also proved, that smaller particles (at the same monomer concentration) are more efficient than bigger ones. The study constitutes a basis for optimization of nanoparticles for their future utilization in in vivo imaging, as nanoparticle size and surface modifications represent crucial parameters for specific targeting, such as for tumors and subsequent photothermal therapy.

## Figures and Tables

**Figure 1 nanomaterials-11-02457-f001:**
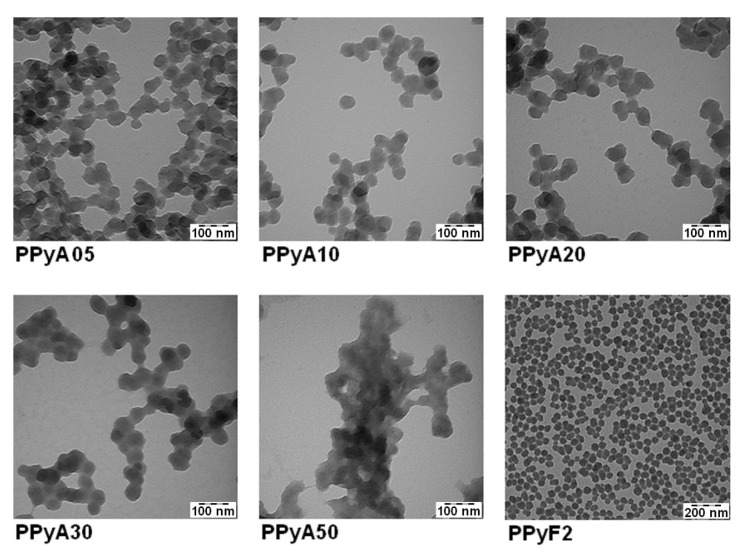
Transmission electron microscopy (TEM) micrographs of the polypyrrole nanoparticles.

**Figure 2 nanomaterials-11-02457-f002:**
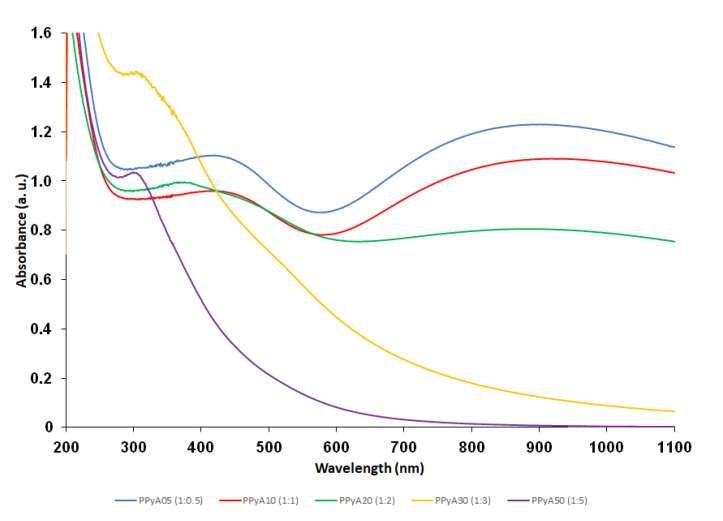
UV-Vis spectra of the polypyrrole nanoparticles prepared by oxidation of Py monomer with different molar ratios of APS (samples PPyA05, PPyA10, PPyA20, PPyA30 and PPyA50) dispersed in physiological PBS (c = 55 µg/mL).

**Figure 3 nanomaterials-11-02457-f003:**
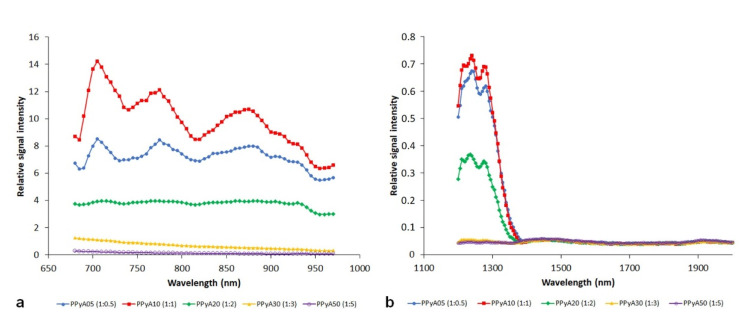
Photoacoustic spectra of polypyrrole nanoparticles prepared with APS oxidant. Different pyrrole and APS ratios are presented at 680–970 nm (**a**) and 1200–2000 nm (**b**) ranges. NP concentration was 3 mg/mL.

**Figure 4 nanomaterials-11-02457-f004:**
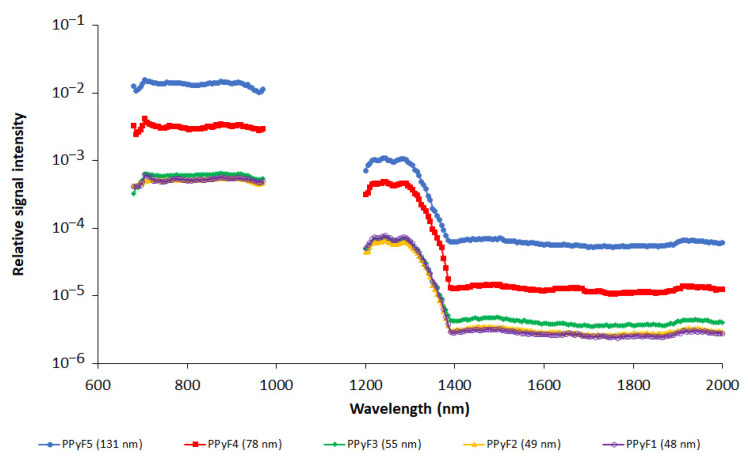
Photoacoustic spectra of polypyrrole nanoparticles prepared with FeCl_3_ oxidant. Different pyrrole and oxidant ratios are presented at 680–970 nm and 1200–2000 nm ranges. Spectra are recalculated per one nanoparticle. Note logarithmic scale on y-axis. NP concentration was 3 mg/mL.

**Figure 5 nanomaterials-11-02457-f005:**
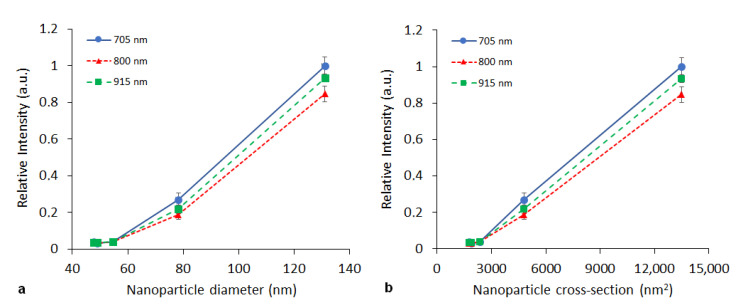
Dependence of the relative PA signal intensity (per one nanoparticle) on the nanoparticle diameter (**a**) or cross-sectional area (**b**) measured at the same concentration (3 mg/mL) at selected wavelengths (705, 800, 915 nm).

**Figure 6 nanomaterials-11-02457-f006:**
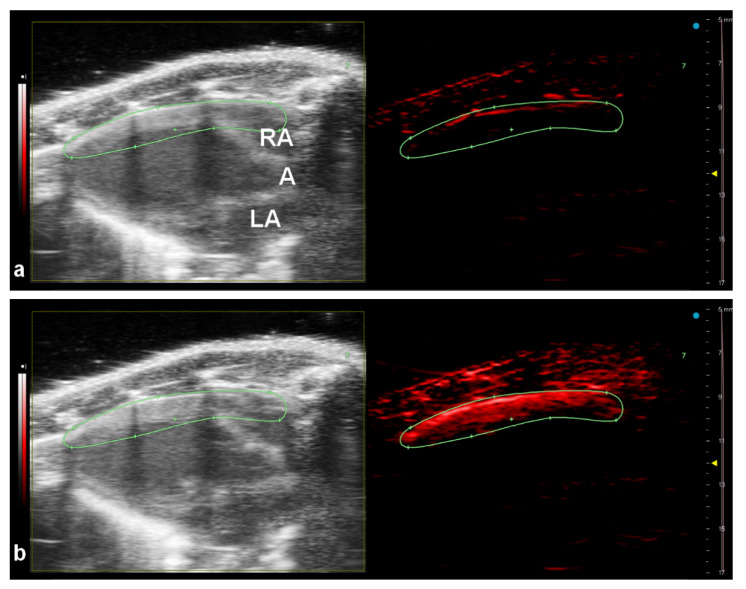
Multimodal US-PA imaging of the mouse heart in its long axis in B-Mode (**left panels**) and PA-Mode (**right panels**) before intravenous PPyF4 nanoparticle administration (**a**) and 5 s after bolus injection (180 µL) (**b**). The PA detection was at single wavelength mode obtained at 800 nm. RA—right atrium, A—aorta, LA—left atrium.

**Table 1 nanomaterials-11-02457-t001:** Properties of the prepared particles: oxidant used for preparation and its ratio, particle diameter (*D*_n_) and dispersity (*Đ*), hydrodynamic diameter (*D*_h_), polydispersity index (*PI*), *ζ*-potential.

Particle	*n*Py:*n*Oxidant	*D*_n_ (nm)	*Đ*	*D*_h_ (nm)	*PI*	*ζ* (mV) *
PPyA05	1:0.5 APS	37.8	1.06	218 ± 4	0.165 ± 0.006	2.2 ± 0.2
PPyA10	1:1 APS	42.3	1.04	302 ± 2	0.214 ± 0.013	1.57 ± 0.09
PPyA20	1:2 APS	43.2	1.05	452 ± 4	0.278 ± 0.012	−0.1 ± 0.7
PPyA30	1:3 APS	50.1	1.11	428 ± 9	0.289 ± 0.160	−4.7 ± 0.8
PPyA50	1:5 APS	44.5	1.09	490 ± 20	0.335 ± 0.027	−8 ± 1
PPyF1	1:2.3 FeCl_3_	47.6	1.15	110.3 ± 0.5	0.149 ± 0.008	28.2 ± 0.7
PPyF2	1:2.3 FeCl_3_	49.1	1.04	112 ± 2	0.160 ± 0.010	21.3 ± 0.6
PPyF3	1:2.3 FeCl_3_	54.6	1.04	126.0 ± 0.8	0.100 ± 0.012	29.2 ± 0.6
PPyF4	1:2.3 FeCl_3_	78.2	1.04	134 ± 1	0.068 ± 0.014	36.5 ± 0.2
PPyF5	1:2.3 FeCl_3_	131.1	1.12	226 ± 2	0.085 ± 0.015	42.5 ± 0.4

* PPyA05-A50 (pH = pH ~3, 25 °C), PPyF1-F5 (pH ~4.5, 25 °C).

## Data Availability

Data supporting reported results can be found in the electronic Appendix A; raw data are available upon request.

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
