# Peer review of "Photoacoustic Properties of Polypyrrole Nanoparticles"

_nanomaterials, 2021, doi:10.3390/nano11092457_

Round 1

Reviewer 1 Report

From my point of view, the article can be accepted after minor revisions. 

First of all, the introduction section should be expanded, mainly with literature overview regarding photacoustic characterization of different types of nanoparticles and applications of photoacoustic approach for imaging.

Also the conclusion section should be improved, the physical conclusions should be presented (not only statements)

For me it is not clear how photoacoustic spectra were measured by authors. What was the configuration of the set-up? What was the sources of parasitic signal and how was it cleared?

How was also measured PA intensity (Fig. 5a)? Why is it nonlinear? How authors may explain this? What was experimental conditions? 

Also, for me it  is better to describe in more details with figure experimental set-up for in-vivo measurements. 

Author Response

REV. 1

We are grateful for thorough work of the reviewers and their valuable comments. Below (in italics) are our comments and changes, which we made according to the reviewer’s notes.

Comments and Suggestions for Authors

From my point of view, the article can be accepted after minor revisions. 

First of all, the introduction section should be expanded, mainly with literature overview regarding photacoustic characterization of different types of nanoparticles and applications of photoacoustic approach for imaging.

The introduction has been expanded; relevant up-to-date references have been added.

Also the conclusion section should be improved, the physical conclusions should be presented (not only statements)

We have rewritten the conclusion section as recommended.

For me it is not clear how photoacoustic spectra were measured by authors. What was the configuration of the set-up? What was the sources of parasitic signal and how was it cleared?

An image illustrating measurement of the spectra has been added to the electronic supplement as well as a photograph showing the experimental setup for in vivo measurements.

How was also measured PA intensity (Fig. 5a)? Why is it nonlinear? How authors may explain this? What was experimental conditions? 

Figure 5 does not represent another measurement; it is just recalculation of the already obtained data shown in Fig. 4. While Fig. 4 shows whole spectra of different samples, Fig. 5 shows relative intensities at three selected excitation wavelengths only – the data are recalculated per one particle. The dependence on the diameter is non-linear, however, the signal is proportional to the cross-sectional area (or particle surface), which - we suppose – corresponds to the fact that heat absorption is proportional to the particle cross section (Shahbazi, 2019) in the case of small optical skin depth. Therefore, there is a linear dependence on the particle surface, i.e., square of the particle size. Explanation is provided in the main text (Discussion).

Also, for me it  is better to describe in more details with figure experimental set-up for in-vivo measurements. 

A photograph showing our experimental setup for in vivo measurements has been added to the supplement.

Reviewer 2 Report

The paper deals with the synthesis of polypyrrole particles, which can be used for photoacoustic imaging. The paper's aim is precise; the experiment design is given in enough detail, the findings are new and supported with the data.

The paper could be published after some issues are improved and resolved:

  1. The paper title and introduction. In my opinion, "photoacoustic properties" is not a very good term as it is ambiguous, to say the least. Moreover, it somewhat warps or clouds the paper's aim, which is much clearer: to make a synthesis procedure, test the optical and thermophysical properties of the synthesized products, and test the products in generic photoacoustic imaging. Thus, the title may be improved by 'synchronizing it with the paper's introduction and aims. Also, I suggest changing the "photoacoustic properties" throughout the text.
  2. The introduction. Also, the aim should be clearly and unambiguously written at the end of the introduction. Now it is ended with some details, which require either a paper aiming or transferring into the beginning of the Results section.
  3. The introduction can also be improved by removing the apparent phrases on the photoacoustic effect (Ref. 4 is certainly not needed here). Also, the focus of the introduction could be changed: in fact, the main subject is the polypyrrole particles with the specified properties. Proving their production and properties is the paper's main novelty, while photoacoustic imaging is more like a recipient of these properties. Thus, the part of the Introduction on the relevance of photoacoustic imaging could be more strict and compact, the necessity of new products for the imaging more explicit, and the short discussion on the polypyrrole and neighboring nanomaterials more detailed.
  4. References. In my opinion, the necessary changes according to the items above will provide changes in the literature as well. However, I would like to mention that a paper of so a fluently developing area as photoacoustic imaging in the middle of 2021 should have papers on the most recent advances in photoacoustic imaging and similar materials.
  5. References. I believe that giving the references mainly in the introduction is not entirely correct, and I encourage authors to compare their findings with the existing data and the references.
  6. Conclusions. In the present form, the conclusions are pretty formal phrases. Probably some outlooks, other features like a wider circle of applications of the product, etc., may be briefly mentioned here.
  7. Data representation. The authors should check the number of significant digits (primarily, Table 1) because, in many instances, the number of significant digits should be 1, not 2 or 3. Moreover, as this Table seems to have confidence intervals, the necessary information (n, P) should be provided in the caption.
  8. Discussion. The nonlinearity of plots in Fig. 5, especially as the data points are not well distributed, should be discussed at least briefly.

Author Response

REV. 2

We are grateful for thorough work of the reviewers and their valuable comments. Below (in italics) are our comments and changes, which we made at the suggestion of the reviewers.

Comments and Suggestions for Authors

The paper deals with the synthesis of polypyrrole particles, which can be used for photoacoustic imaging. The paper's aim is precise; the experiment design is given in enough detail, the findings are new and supported with the data.

The paper could be published after some issues are improved and resolved:

  1. The paper title and introduction. In my opinion, "photoacoustic properties" is not a very good term as it is ambiguous, to say the least. Moreover, it somewhat warps or clouds the paper's aim, which is much clearer: to make a synthesis procedure, test the optical and thermophysical properties of the synthesized products, and test the products in generic photoacoustic imaging. Thus, the title may be improved by 'synchronizing it with the paper's introduction and aims. Also, I suggest changing the "photoacoustic properties" throughout the text.

We are not sure whether it is possible to change the paper title at this stage. If yes, we suggest:

Effect of synthesis and material structure on photoacoustic signal of polypyrrole nanoparticles

  1. The introduction. Also, the aim should be clearly and unambiguously written at the end of the introduction. Now it is ended with some details, which require either a paper aiming or transferring into the beginning of the Results section.

The aim of the study has been reformulated:

Our aim was to test several synthesis procedures for preparation of polypyrrole (PPy) nanoparticles and to study the influence of modifications in polymer preparation on their optical and thermophysical properties in vitro and, as a proof of principle, to test selected nanoparticles as a contrast agent also in vivo, to demonstrate their unique properties for preclinical, or even clinical photoacoustic imaging.

  1. The introduction can also be improved by removing the apparent phrases on the photoacoustic effect (Ref. 4 is certainly not needed here). Also, the focus of the introduction could be changed: in fact, the main subject is the polypyrrole particles with the specified properties. Proving their production and properties is the paper's main novelty, while photoacoustic imaging is more like a recipient of these properties. Thus, the part of the Introduction on the relevance of photoacoustic imaging could be more strict and compact, the necessity of new products for the imaging more explicit, and the short discussion on the polypyrrole and neighboring nanomaterials more detailed.

The Introduction has been rewritten and we have emphasized the role of polypyrrole in photoacoustic imaging. Consequently, references have been updated.

  1. References. In my opinion, the necessary changes according to the items above will provide changes in the literature as well. However, I would like to mention that a paper of so a fluently developing area as photoacoustic imaging in the middle of 2021 should have papers on the most recent advances in photoacoustic imaging and similar materials.

Up-to-date references have been supplemented.

  1. References. I believe that giving the references mainly in the introduction is not entirely correct, and I encourage authors to compare their findings with the existing data and the references.

References have been updated.

  1. Conclusions. In the present form, the conclusions are pretty formal phrases. Probably some outlooks, other features like a wider circle of applications of the product, etc., may be briefly mentioned here.

The section has been reformulated.

  1. Data representation. The authors should check the number of significant digits (primarily, Table 1) because, in many instances, the number of significant digits should be 1, not 2 or 3. Moreover, as this Table seems to have confidence intervals, the necessary information (n, P) should be provided in the caption.

The data have been corrected according to the STD rounded to 1 digit, number of measurement has been supplemented in Methods. P-value is not applicable.

  1. Discussion. The nonlinearity of plots in Fig. 5, especially as the data points are not well distributed, should be discussed at least briefly.

As was explained above, the dependence on the diameter is non-linear, however, the signal is proportional to the cross-sectional area (or particle surface), which - we suppose – corresponds to the fact that heat absorption is proportional to the particle cross section (Shahbazi, 2019) in the case of small optical skin depth. Therefore, there is a linear dependence on the particle surface, i.e., square of the particle size. Explanation is provided in the main text (Discussion).

Reviewer 3 Report

The authors investigated photoacoustic properties of polypyrrole nanoparticles. I think there is no proper motivation for this study and they use many expression of "may be". The studies of "photoacoustic properties of polypyrrole nanoparticles" already published in many journals which they mentioned in the references 20~25, they should address that what new things of this work is. Abstract 1. Better to add some comments for investigating motivation for photoacoustic properties of polypyrrole nanoparticles in abstract. especially, the expression "We investigated properties of polypyrrole nanoparticles, which may considerably enhance contrast in photoacoustic images" is ambiguous. Introduction 2. "The absorbed energy may be transformed into a chemical energy, reemitted, or transformed into heat energy, which causes thermal expansion and the subsequent formation of acoustic (sound) waves." - Why do the authors use the expression of "may be"? The authors should discuss about this expression clearly with adding reference. 3. Special attention - this is not the motivation of this study. The authors should add some comments for the motivation of this study about polypyrrole (PPy) nanoparticles. Main article 4. In 2.3. In vitro photoacoustic imaging - "Suspensions of PPyA samples were diluted to concentrations 1, 2, 3, 4, 6, 8 mg/mL)" -> "Suspensions of PPyA samples were diluted to concentrations (1, 2, 3, 4, 6, 8 mg/mL)" 5. and Raman spectra (data not shown) - This should be removed. 6. Figure 1: Transmission electron microscopy (TEM) micrographs of the polypyrrole 261 nanoparticles.Figure 2: UV-Vis spectra of the polypyrrole nanoparticles prepared by .. -> Figure 1: Transmission electron microscopy (TEM) micrographs of the polypyrrole 261 nanoparticles. Figure 2: UV-Vis spectra of the polypyrrole nanoparticles prepared by.. 7. 680 – 970 nm, 1200 – 2000 nm - where is the 970 nm - 1200 nm results? They should explain about the absence of the wavelength range. 8. In 3.4. In vivo imaging - they should provide reference value of photoacoustic imaging with other materials for comparing with polypyrrole nanoparticles performance (enhancement factor). 9. Photoacoustic imaging represents an emerging imaging technique with so-far unknown possibilities. - Check this expression is firmly right and they should explain it clearly.

Author Response

REV. 3

Thank you for your valuable comments. Below (in italics) are our comments and changes, which we made according to the notes.

Comments and Suggestions for Authors

The authors investigated photoacoustic properties of polypyrrole nanoparticles. I think there is no proper motivation for this study and they use many expression of "may be". The studies of "photoacoustic properties of polypyrrole nanoparticles" already published in many journals which they mentioned in the references 20~25, they should address that what new things of this work is.

Abstract

  1. Better to add some comments for investigating motivation for photoacoustic properties of polypyrrole nanoparticles in abstract. especially, the expression "We investigated properties of polypyrrole nanoparticles, which may considerably enhance contrast in photoacoustic images" is ambiguous.

Modifications were introduced as suggested.

Introduction

  1. "The absorbed energy may be transformed into a chemical energy, reemitted, or transformed into heat energy, which causes thermal expansion and the subsequent formation of acoustic (sound) waves." - Why do the authors use the expression of "may be"? The authors should discuss about this expression clearly with adding reference.

Introduction has been modified, expanded and relevant references have been added.

  1. Special attention - this is not the motivation of this study. The authors should add some comments for the motivation of this study about polypyrrole (PPy) nanoparticles.

Aim of the study has been described in the end of Introduction.

Main article

  1. In 2.3. In vitro photoacoustic imaging - "Suspensions of PPyA samples were diluted to concentrations 1, 2, 3, 4, 6, 8 mg/mL)" -> "Suspensions of PPyA samples were diluted to concentrations (1, 2, 3, 4, 6, 8 mg/mL)"

Corrected.

  1. and Raman spectra (data not shown) - This should be removed.

Removed.

  1. Figure 1: Transmission electron microscopy (TEM) micrographs of the polypyrrole 261 nanoparticles.Figure 2: UV-Vis spectra of the polypyrrole nanoparticles prepared by .. -> Figure 1: Transmission electron microscopy (TEM) micrographs of the polypyrrole 261 nanoparticles.

Figure 2: UV-Vis spectra of the polypyrrole nanoparticles prepared by..

Typesetting corrected.

  1. 680 – 970 nm, 1200 – 2000 nm - where is the 970 nm - 1200 nm results? They should explain about the absence of the wavelength range.

The instrument we used in the study had tunable lasers in the two listed bands. Unfortunately, it does not enable excitation between 970 nm and 1200 nm. Explanation has been added into the text.

  1. In 3.4. In vivo imaging - they should provide reference value of photoacoustic imaging with other materials for comparing with polypyrrole nanoparticles performance (enhancement factor).

An experiment comparing PPyF4 with a commercial ICG has been performed and the results were summarized in the paper.

  1. Photoacoustic imaging represents an emerging imaging technique with so-far unknown possibilities. - Check this expression is firmly right and they should explain it clearly.

The sentence has been reformulated.

Reviewer 4 Report

The study, “Photoacoustic properties of polypyrrole nanoparticles”, by Kesa, el al. has described an interesting contrast agent for photoacoustic imaging. They tested the polypyrrole nanoparticles at different type via in vitro and in vivo studies. Even though, the studies using polypyrrole nanoparticles for photoacoustic imaging have been published, this study shows the results from the different agent type.

Overall, the study has valuable impact for introducing the PA nanoparticle. However, it would be better with modification of the manuscript.

  • “Although any electromagnetic radiation 39 may induce PA effect, most applications require radiation in the ultraviolet to infrared 40 wavelength range.” (P1 39 40) This needs references.
  • In the experiments, how much was the laser power?
  • Line 219, ‘Subtraction Control function’ was used for in vivo study? In Results, what image processing was done for figure 6?
  • In Figure 6 legend, please indicate the amount of bolus injection.
  • Figure 6 a and b, the two PA images are using the same color map? Do they show the images in the same PA intensity range?
  • Please indicate the aorta and atriums in figure 6.
  • Please indicate the injection time on Fig. S2, if the injection is not done at 00:00 on x-axis. If the max is at 5 seconds after injection, the zero on time line should not be the injection time.
  • Why choose PPYF4 for in vivo study?
  • Why use 800 nm for imaging?
  • What application or utility will be available with the nanoparticle? Currently, it shows heart structure imaging only.

Minor

  • In the manuscript, ‘In vivo’ should be
  • 109 ‘(‘ is missing.
  • 128, PLL is poly-L-lysine?
  • The line for Figure 2 legend is wrong.
  • 316, ‘The intense photoacoustic signal was clearly visible in the anterior 315 wall of the mouse heart immediately after NPs injection (Fig. 6a).’ is it correct? Fig. 6a or 6b? The sentence explains about the PA signal after injection.
  • Figure 6 legend, last sentence could be changed like ‘the PA detection at single wavelength mode was obtained at 800nm.

Author Response

REV. 4

We are grateful for the valuable comments. Below (in italics) are our comments and changes, which we made according to the reviewer’s notes.

Comments and Suggestions for Authors

The study, “Photoacoustic properties of polypyrrole nanoparticles”, by Kesa, el al. has described an interesting contrast agent for photoacoustic imaging. They tested the polypyrrole nanoparticles at different type via in vitro and in vivo studies. Even though, the studies using polypyrrole nanoparticles for photoacoustic imaging have been published, this study shows the results from the different agent type.

Overall, the study has valuable impact for introducing the PA nanoparticle. However, it would be better with modification of the manuscript.

  • “Although any electromagnetic radiation 39 may induce PA effect, most applications require radiation in the ultraviolet to infrared 40 wavelength range.” (P1 39 40) This needs references.

A reference (Razansky D. 4.19 - Optoacoustic Imaging, p. 281-300, in Comprehensive Biomedical Physics, Editor: Anders Brahme, Elsevier, 2014, ISBN 9780444536334, https://doi.org/10.1016/B978-0-444-53632-7.00421-4) has been added.

  • In the experiments, how much was the laser power?

Laser pulse energy was 36 mJ for in vitro spectra acquisition, 30 mJ, pulse rate 20 Hz, pulse width < 10 ns, for in vivo measurements at 800 nm. The information has been added to the Methods.

  • Line 219, ‘Subtraction Control function’ was used for in vivo study? In Results, what image processing was done for figure 6?

Subtraction Control function was used for post-processing in the in vivo study to decrease the signal originated from hemoglobin, as stated in the Methods. No further post processing was used for data in Fig. 6.

  • In Figure 6 legend, please indicate the amount of bolus injection.

The amount has been added to the legend.

  • Figure 6 a and b, the two PA images are using the same color map? Do they show the images in the same PA intensity range?

Yes, same colormap was used for both images and they show same intensity range.

  • Please indicate the aorta and atriums in figure 6.

We have added description to the Figure 6.

  • Please indicate the injection time on Fig. S2, if the injection is not done at 00:00 on x-axis. If the max is at 5 seconds after injection, the zero on time line should not be the injection time.

The bolus was injected at time t = 0 s, the figure was modified to make it clear.

  •  
  • Why choose PPYF4 for in vivo study?

A medium-sized nanoparticles (78 nm) have been chosen for the in vivo experiment. Nanoparticles of this size are usually well tolerated in vivo and also easily excreted by the renal pathway and thus do not represent a significant long-term burden for the organism.

  • Why use 800 nm for imaging?

We used 800 nm to minimize nature background.

  • What application or utility will be available with the nanoparticle? Currently, it shows heart structure imaging only.

Several potential proposals have been added to Discussion/Conclusion sections.

Minor

  • In the manuscript, ‘In vivo’ should be
  • 109 ‘(‘ is missing.
  • 128, PLL is poly-L-lysine?
  • The line for Figure 2 legend is wrong.
  • 316, ‘The intense photoacoustic signal was clearly visible in the anterior 315 wall of the mouse heart immediately after NPs injection (Fig. 6a).’ is it correct? Fig. 6a or 6b? The sentence explains about the PA signal after injection.
  • Figure 6 legend, last sentence could be changed like ‘the PA detection at single wavelength mode was obtained at 800nm.

All the points have been corrected. Thank you for thorough reading.

Round 2

Reviewer 3 Report

Thank you for the revision. I recommend to be published.